# Make a choice: A rapid strategy for minimizing peat in horticultural press pots substrates using a constrained mixture design and surface response approach

André Sradnick👤*, Marie Werner, Oliver Körner👤

Leibniz Institute for Vegetable and Ornamental Crops (IGZ) e.V., Grossbeeren, Germany

* sradnick@igzev.de

**Data Availability Statement:** All relevant data are within the manuscript and its Supporting information files.

## Abstract

Peat is the most common used substrate in horticultural seedling production. To reduce peat in horticultural potted plant cultivation systems in general is an obstacle, even within the highly specialized horticultural industry. Next to soil-less cultivation systems as e.g. hydroponics, the horticultural industry is eagerly looking for suitable peat substitutes. The demands on these compounds are high, basically mimicking the physical properties of peat. A 100% replacement of peat for press-pots used in seedling production has not yet been found, and only mixes of peat and substrates exist. Several suitable peat substitutes with different properties are known, that usually are used as a share of a mixed peat-substitute substrate. A constrained mixture design was used to test substrates containing 50% v/v and 25% v/v peat and four peat substitutes (two composts and two wood fibers) for vegetable seedling production. By limiting the maximum quantities of each material to be added, there was no negative effect on the growth of Chinese cabbage (*Brassica rapa subsp. pekinensis*). This means a reduction in of peat to 25% v/v is possible without a change in substrate quality. The mixture design allowed a quick decision to be made regarding the most suitable peat-reduced mixtures. The surface response approach enabled the experimental results to be easily transferred to horticultural practices, additionally. This flexible and efficient method also allows the predictions to be used to meet specific crop management needs.

## Introduction

Currently, approximately 38% of peat produced in Europe is used for non-energy purposes [1]. The horticultural sector accounts for a significant portion of this usage, at approximately 65% [2]. Peatlands are essential natural carbon sinks due to their capability of storing substantial amounts of carbon dioxide ($CO_2$-C). However, when peat is used as a seedling substrate in nurseries, additional $CO_2$-C is emitted as the organic matter is mineralized at a rate of 5% per year [1] and thus contributes to the input of greenhouse gases into the atmosphere [3]. Furthermore, the peat extraction process disrupts the peatland ecosystem in a sustainable manner [4].

**Funding:** Funding for the research was provided by the German Federal Ministry of Food and Agriculture (BMEL) through the project "Development and evaluation of peat-reduced production systems in horticulture (ToPGa)" (project 2220MT006C). The funders had no role in study design, data collection and analysis, decision to publish, or preparation of the manuscript.

**Competing interests:** The authors have declared that no competing interests exist.

The challenge of reducing the use of peat in European horticulture has been acknowledged, with many countries ratifying sustainable peat reduction targets [1]. This implies that peat-based soilless systems, commonly employed in horticulture, may need to be re-evaluated in terms of their water supply and nutrient management [5]. Moreover, it has been projected that the demand for substrates may significantly increase in the coming decades [6].

In order to address the aforementioned challenges, peat-reduced or peat-free substrates must continue to provide the necessary technological requirements to be compatible with highly automated horticultural companies. This is particularly pertinent to ensure consistent substrate quality and meet the specific requirements for various horticultural sectors such as ornamental plants, herbs, woody plants, soft fruit and nursery plants.

Peat is commonly replaced with regenerative organic materials such as compost, fermentation residues, wood fibers, fiber plants, sphagnum moss, or coconut fiber [7–10]. However, may not be suitable as sole peat substitutes, either because their inability to meet physical, biological, and chemical requirements, or because they cannot meet practical demands in terms of availability. In the horticultural industry, mixtures are utilized for the purpose of achieving desired qualities [5].

Compost can be an important replacement for peat due to its wide availability. During composting, organic material is transformed microbiologically through aerobic processes, resulting in an improvement in several of its properties, making it a suitable substrate [11]. Green compost is highly compatible due to its low salt content compared to other composts [12]. However, compost may have often a pH above 7, which limits its use as a peat substitute aggregate.

Wood fiber has been used in multiple sources as an alternative to peat [13–15]. Similarly to compost, it is currently obtainable in sufficient quantities. The use of fibrous materials derived from wood or fibrous plants is suitable as a peat substitute, particularly due to its low salt concentration and advantageous hydrological characteristics. Nonetheless, its usage is constrained by high nitrogen immobilization potential [13].

Research into potential substitutes for peat has been ongoing for decades [16]. The majority of research into peat alternatives does not explore whether peat can be replaced, but instead what percentage, what proportions and what materials are suitable for substitution [17–20]. Various sources suggest that a mixture of peat substitutes may be a viable alternative [21]. However, constructing a substrate mixture to reduce the amount of peat used can be challenging since each substitute can impact the properties of the mixture differently; e.g. change physical properties like maximum water holding capacity ($WHC_{max}$) or stability against pressing, bio chemical properties like pH and EC or biological properties like N-Immobilization or pathogens.

Promising approaches have been developed to decrease the amount of peat in traditional mixtures through the combination of peat alternatives [21, 22]. Particularly, the surface response analysis approach appears to be promising [7]. This method has great potential for practical decision making and is easily transferable to horticultural industry by a surface response approach. Decisions can be made swiftly with a minimal amount of experimentation by employing designs for formulation (mixture designs) based on the simplex-centroid or constrained mixture approach [23].

According to Pascual et al. [7] the direct transfer of experimental results to practice should be met through experiments involving mixtures. Those need to take the unique characteristics of mixtures into account [23]. In the case of press pots for seedling cultivation, the simplex centroid approach proposed by Ceglie et al. [21] recommending mixtures with a maximum peat-substitute content of 90% is likely to be ineffective due to physical constraints (press ability and physical stability). These constraints may be limited in horticultural practice. An alternative design was proposed by several authors how descripted an approach based constrained

mixture design [24–26]. In this approach, only a subarea of the potential mixture combinations is considered. The application of a region-defining system for decision-making, which has been utilized in various fields [26–29], has enabled the consideration of previously excluded materials in lower admixtures. To further enhance the decision-making process, response surface method combined with desirability should be employed to support the experimental design.

The purpose of this study was to evaluate whether an efficient decision-making process can be developed by establishing threshold levels for mixing compost and fibrous materials with a comprehensive mixing design with predetermined regions. This research attempted to generate peat-reduced substrates that meet the actual requirements. The hypothesis generated from this study is that by establishing threshold levels, the negative impact on plant growth can be minimized as unfavorable physical effects can be prevented from the beginning. This allows for a more effective "fine-tuning" of the mixture to be conducted within the "defined regions", and it facilitates the transfer of results and predictions to help decrease the use of peat further. We have conducted two experiments under practice-like conditions to reduce peat in by mixing several peat substitutions. We conclude that the flexible and efficient approach supports the horticulture industry to reduce peat.

## Material and methods

### Experimental conditions

In March/April 2022 and September/October 2022, two experiments were conducted at the Leibniz-Institute of Vegetable and Ornamental crops (Grossbeeren, Germany; LAT 52°) to investigate the growth of Chinese cabbage (*Brassica rapa subsp. pekinensis*, cultivar: 'Granaat') under standard practice conditions with different peat mixtures were used with 50% v/v or 25% v/v peat in Experiment (Exp.) 1 or 2, respectively. The peat substitutes chosen were Terraktiv® green compost (GC), TerrAktiv FT® fermented compost fiber (FC), GreenFibre® fine wood fiber soft (SF, Klasmann Deilmann GmbH, Germany) and LignoDrain® wood fiber coarse (RF, Kekkilä-Brill Substrates GmbH & Co. KG, Germany). The four mixes were benchmarked against a set amount of Potgrond P, black peat (control (C); Klasmann-Deilmann GmbH, Germany). All materials were supplied by the respective companies and were subsequently analyzed according to the standards of the Association of German Agricultural Analytic and Research Institutes (VDLUFA) (Table 1).

250 seedlings were propagated from seed of each test mixture (M) and the control (sown at 26th March 2022 and at 24th September 2022 in Exp. 1 and 2, respectively). After germination the seedlings were grown for 21 days in a climate-controlled greenhouse (heating and ventilation temperature set point of 18 and 22 day and night, respectively) on benches, followed by a one week period in open air. All seedling were grown in standard plant raising trays (624 mm x 425 mm x 110 mm). Temperature curves for the experiments are provided in the supplementary material (S1 and S2 Figs). The mixtures were rewetted using the potting density method described by Raviv et al. [30] and adjusted to optimum moisture content by hand approximately one week prior to use, and then repeated every 2 to 3 days as needed. The press pots (4.0 cm x 4.0 cm, w l) were made with a potting machine (Unger, Perfekt, Dossenheim, Germany). The heights of the press pots were slightly different in the two experiments, resulting in different press pot volumes (S1 Table).

In both experiments, treatment against the mourning fly occurred after 10 days with the *Steinernema feltiae* preparation (nemaplus®). For the second Exp., 10 mg N of ammonium nitrate was added to each pot via pipette, with the exception of the control, in order to avoid nitrogen deficiency.

**Table 1. Properties of the constituents for growing media in Exp. 1 (50% peat v/v) and Exp. 2 (25% peat v/v).**

| | Potting volume | | N | C/N | pH | Salt | $N_{min}$-N | $NH_4$-N | $NO_3$-N | P | K | Mg | Na |
|---|---|---|---|---|---|---|---|---|---|---|---|---|---|
| | g FM L$^{-1}$ | g DM L$^{-1}$ | g kg$^{-1}$ | | | | mg g$^{-1}$ DM | | | | | | |
| | Exp. 1 50% v/v peat | | | | | | | | | | | | |
| **Green compost (GC)** | 793 | 555 | 8.05 | 15 | 6.8 | 4.71 | 0.34 | 0.01 | 0.33 | 0.26 | 3.10 | 0.46 | 0.23 |
| **Fermented compost (FC)** | 430 | 241 | 9.48 | 27 | 5.9 | 12.74 | 2.53 | 0.28 | 2.25 | 0.14 | 1.55 | 0.50 | 0.16 |
| **Wood fiber soft (SF)** | 306 | 87 | 0.63 | 746 | 4.3 | 1.02 | 0.03 | 0.02 | 0.02 | 0.05 | 0.87 | 0.20 | 0.06 |
| **Wood fiber raw (RF)** | 230 | 53 | 0.53 | 885 | 4.7 | 1.08 | 0.08 | - | - | - | 1.09 | 0.15 | - |
| **Peat** | 582 | 146 | 11.68 | 41 | 5.6 | 9.55 | 1.74 | 1.08 | 0.66 | 0.42 | 1.44 | 1.68 | 0.32 |
| | Exp. 2 25% v/v peat | | | | | | | | | | | | |
| **Green compost (GC)** | 793 | 448 | 11.6 | 16 | 6.9 | 5.79 | 0.29 | 0.01 | 0.28 | 0.32 | 4.43 | 0.57 | 0.28 |
| **Fermented compost (FC)** | 453 | 215 | 10.5 | 31 | 6.2 | 10.77 | 2.33 | 0.33 | 2.00 | 0.16 | 1.62 | 0.46 | 0.17 |
| **Wood fiber soft (SF)** | 176 | 81 | 0.65 | 760 | 4.9 | 0.69 | 0.01 | 0.01 | 0.01 | 0.03 | 0.55 | 0.14 | 0.03 |
| **Wood fiber raw (RF)** | 150 | 55 | 0.49 | 1022 | 5.1 | 0.58 | 0.02 | - | - | - | 0.32 | 0.20 | - |
| **Peat** | 503 | 140 | 11.07 | 44 | 5.7 | 6.95 | 1.14 | 1.05 | 0.09 | 0.33 | 1.09 | 1.58 | 0.30 |

## Measurements

The concentration of $N_{min}$-N was determined in frozen samples (-20°C) that had been submerged in a 0.0125 M CaCl$_2$ solution. The total press pot volume was measured three times, four days after seeding (DaS), 17 and 18 DaS, or 27 and 28 DaS (end of experiment) for Exp. 1 and Exp. 2, respectively. The press pot density was calculated after drying for 24 hours at 105°C in a drying oven. The pH, salinity and maximum water holding capacity of the press pots were assessed at four DaS and at the end of the experiment. The pots were set to a water holding capacity of 50–60%. The force required to cause the pots to collapse was measured using a Sauter FA 200 Force gauge 200 N and recorded in kg cm$^{-1}$. The plant biomass was determined in terms of fresh mass using a laboratory balance after 17 and 18 DaS for Exp. 1 and Exp. 2, respectively, and at the end of the trial. The mean value of 3 (17, 18 DaS) or 10 plants (end of experiment) was calculated.

## Experimental design

The constrained mixture (extreme vertices) design and centroids approach, "XVERT," was utilized to create an experimental design for the tests described by Smith [23]. XVERT is an advanced iteration of the M-A algorithm, offering increased efficiency and versatility by allowing for the generation of either all or a fraction of the vertices [23, 31]. A design incorporating five components was employed, while maintaining a constant proportion of peat across the experimental conditions. The purpose of this design was to predict mixing effects with a minimum number of test mixtures while considering the respective limits of the four peat alternatives. The mixture design was created using the R package "mixexp" [32, 33].

For the Exp 1, which had a constant share of 50% v/v peat, the maximum limits of GC and FC were set at 35% v/v due to the composted material and high salt content of FC. For SF and RF, the maximum limit was set at 15% v/v in order to avoid potential high N immobilization. Taking into account the specified limits, the XVERT algorithm proposed a design matrix consisting of seven mixtures (Table 2). For Exp. 2 (25% peat mix), the limits for GC, FC, SF, and RF were set at 10–40%, 25%, 35%, and 10%, respectively, based on the predictions from Exp. 1. The limit for the sum of SF and RF was 40% v/v. Table 3 provide an overview of the test mixtures for both experiments. The mixture design was created using the R package "mixexp"

**Table 2. Component bounds (design constraints) for the mixture matrix calculation in % v/v.**

|  | Lower limit | Upper limit |
|---|---|---|
|  | Exp. 1: 50% peat | |
| GC | 0 | 35 |
| FC | 0 | 35 |
| SF | 0 | 15 |
| RF | 0 | 15 |
| SF+RF | 0 | 15 |
|  | Exp. 2: 25% peat | |
| GC | 10 | 40 |
| FC | 0 | 25 |
| SF | 0 | 35 |
| RF | 0 | 10 |
| SF+RF | 0 | 40 |

[32, 33]. The experiments were conducted in a completely randomized block design with five replicates, with one seedling as experimental unit. Samples were taken from the middle part of the press pot racks and then replaced by resetting.

## Statistics and calculations

The immobilization of nitrogen was calculated by taking the mean of the first sampling and subtracting the inorganic nitrogen from the second and third sampling, respectively. Statistical

**Table 3. Experimental matrix of the constrained mixture design for Exp. 1 and Exp. 2.**

| Mixture | GC | FC | SF | RF | Peat |
|---|---|---|---|---|---|
|  | Vol [%] | | | | |
|  | Exp. 1: 50% peat | | | | |
| M 1 | 35 | 15 | 0 | 0 | 50 |
| M 2 | 35 | 0 | 15 | 0 | 50 |
| M 3 | 35 | 0 | 0 | 15 | 50 |
| M 4 | 0 | 35 | 15 | 0 | 50 |
| M 5 | 0 | 35 | 0 | 15 | 50 |
| M 6 | 15 | 35 | 0 | 0 | 50 |
| M 7 | 20 | 20 | 5 | 5 | 50 |
| C | 0 | 0 | 0 | 0 | 100 |
|  | Exp. 2: 25% peat | | | | |
| M 1 | 40 | 0 | 35 | 0 | 25 |
| M 2 | 40 | 25 | 10 | 0 | 25 |
| M 3 | 40 | 25 | 0 | 10 | 25 |
| M 4 | 10 | 25 | 35 | 5 | 25 |
| M 5 | 15 | 25 | 35 | 0 | 25 |
| M 6 | 40 | 0 | 25 | 10 | 25 |
| M 7 | 10 | 25 | 30 | 10 | 25 |
| M 8 | 35 | 0 | 35 | 5 | 25 |
| M 9 | 35 | 0 | 30 | 10 | 25 |
| M 10 | 29 | 14 | 26 | 6 | 25 |
| C | 0 | 0 | 0 | 0 | 100 |

analysis was carried out using R software, with a linear model as described in Lawson and Willden [33]. A Scheffé linear model for mixtures was used, under the assumption of homogeneity of variance using "Levene's Test" from R "car" package [34].

$$\hat{y} = \sum_{i=1}^{q} \beta_i x_i \tag{1}$$

Moreover, we utilized the "Shapiro-Wilk test" from the "stats" package in R to assess the normal distribution of the linear model. Additionally, the "Durbin-Watson test" from the "car" package in R was employed to examine autocorrelation. The results indicated that no data transformation was required for the datasets.

In order to assess the variations among the test mixture, an analysis of variance (ANOVA) followed by a Tukey's honest significant difference (HSD) test was conducted utilizing the "stats" package in R. The critical distance (CD), which serves to quantify significant differences, was derived by dividing the range of the confidence interval by two.

The validity of the models was evaluated using root mean squared error (RMSE), corrected multiple R-squared ($R^2$) and RPD. RPD is the ratio of the standard deviation of the observations and the standard error of prediction. The following criteria were used to assess prediction accuracy: excellent prediction (RPD > 3.0 and $R^2$ > 0.9), good prediction (RPD from 2.5 to 3.0 and $R^2$ from 0.82 to 0.90), approximate quantitative prediction (RPD from 2.0 to 2.5 and $R^2$ 0.66 to 0.81), distinguish between high and low values (RPD from 1.5 to 2.0 and $R^2$ 0.5 to 0.65) and failed prediction for values lower than RPD 1.5 and $R^2$ 0.50.

The overall desirability of the surface response was determined by the R package 'desirability' based on Derringer and Suich [35]. The desirability function transforms estimated response variables, into desirability values. These values range between 0 and 1, with higher values indicating greater desirability. By combining this individual desirability using the geometric mean, an overall assessment of the combined response levels was obtained. The values have the interval (0 to 1) and increases as the properties' balance becomes more favorable. Notably, if any individual desirability is 0, indicating an unacceptable response variable, then the overall desirability was also 0, indicating an unacceptable product. The choice of the geometric mean is driven by these properties and considerations [35]. The properties used to calculate the individual and overall desirability are listed in Table 4.

Representation of the boundaries and decisions using a mixing triangle was done using the 'mixexp' package's 'modelplot' function. For the illustration of effects on predicted properties, effect plots based on "Piepels direction" were estimated using the 'ModelEff' function of the "mixexp" package. These effect plots can be found in the Figures of the supplementary materials (S3 and S4 Figs). These plots provide visual representations of the effects of different mixture components on the predicted properties. The scheme of the test procedure is illustrated in Fig 1.

**Table 4. Desirability criteria with, target and min and max boundary for Exp. 1 (50% peat v/v) and Exp. 2 (25% peat v/v).** Days after seeding (DaS), minimum decision value (Min), maximum decision value (Max).

| Parameter | Decision | Min | Max | Target | Unit |
|---|---|---|---|---|---|
| pH (DaS 4) | target | 5.5 | 7 | 6.5 | |
| Salt content (DaS 4) | min | 0 | 200 | | mg pot$^{-1}$ |
| Nmin-N reduction (End of Exp.) | min | -5 | 20(Exp1)/25(Exp2) | | mg pot$^{-1}$ |
| WHCmax (DaS 4) | max | 45 | 65(Exp1)/70(Exp2) | | g |
| Ph. Stability (End of Exp.) | max | 0.55(Exp1)/0.40(Exp2) | 1 | | kg cm$^2$ |
| Density (DaS 4) | min | 0.1 | 0.65 | | g cm$^3$ |

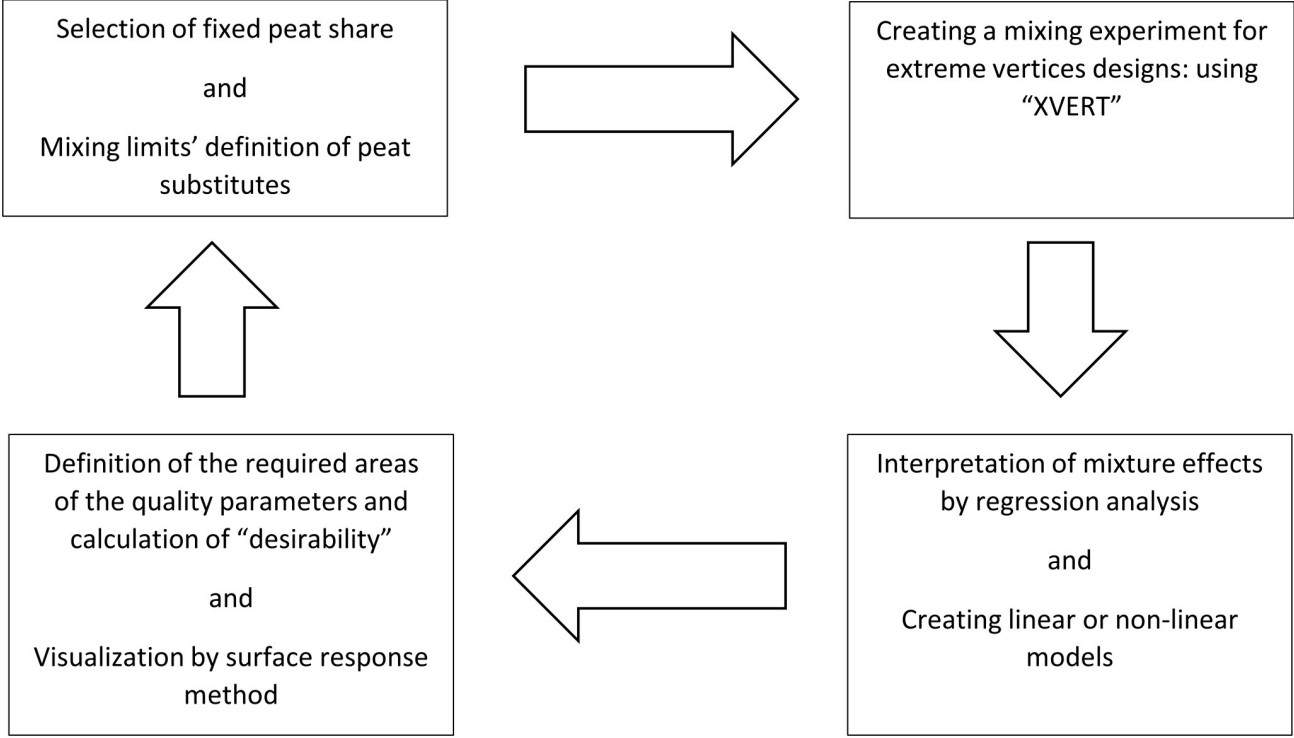

**Fig 1. Description of the methical procedure for handling peat reduction by using mixture designs and the surface response method.**

## Results

Growing media constituents For Exp. 1 (50% v/v Peat) and Exp. 2 (25% v/v Peat), new batches were used. Table 1 indicates that the qualities of the materials only had minor differences. Notably, the FC exceeding 10 mg salt per g (Table 1). The pH value was lower than 7 in all growing media constituents.

Additionally, the inorganic N ($N_{min}$-N) content in the peat and FC was in excess of 1 mg $g^{-1}$ DM. Generally, the nutrient contents in the RF and SF were smaller than 1 mg $g^{-1}$ DM, with C/N ratio > 700 (Table 1).

### Seedling growth

In Experiment 1, the biomass of seedlings treated with a 50% v/v share of peat exhibited a range of 0.70 g FM $pot^{-1}$ (specifically, Mix (M) 6) to 0.88 g FM pot-1 (M 7). In Exp. 2, the biomass ranged between 0.78 g FM $pot^{-1}$ (M 2) and 1.16 g FM $pot^{-1}$ (M4) 18 DaS (S1 Table).

By the end of Exp. 1, the biomass of Chinese cabbage varied from 2.04 g $pot^{-1}$ to 2.82 g $pot^{-1}$, with M1 being 28% lower than the control treatment. In Experiment 2, M10 exhibited a 29% lower biomass compared to M9 (S1 Table), while the overall biomass ranged between 3.02 g $pot^{-1}$ and 4.20 g $pot^{-1}$. However, regression analysis of both experiments revealed no significant effects of the individual substrate components on plant growth, with $R^2$ values < 0.26 and an RPD < 1.2 (S3 Table).

## Physico-chemical, chemical properties physical properties

**pH-value and salt content.**   In Experiment 1, the pH ranged between 5.77 and 6.03. In Exp. 2, with a 25% v/v peat concentration, the pH ranged between 6.58 and 6.80. At the beginning of Exp. 1, the highest pH value was observed in mixtures M1 to M3, which had high GC contents (S1 Table). This trend was supported by the increasing piepels directions of the trace plot observed in Exp. 1 and Exp. 2 (S3 Fig). A reliable prediction of the pH value was possible for the initial sampling date (4 DaS), with an $R^2$ of 0.87 and 0.89, and a prediction error of 0.05 and 0.06, respectively. However, qualitative predictions for the pH value at the last sampling date were not feasible for both experiments, yielding $R^2$ values of 0.54 and 0.38, respectively (S1 Table).

In Exp. 1, the mixtures M1 and M2, which had the lowest wood fiber concentration, exhibited the highest salt content. However, when the wood fiber concentration exceeded 15% v/v, the salinity decreased to a level comparable to the control group. In Experiment 2, mixtures M1-4, with a high compost concentration, displayed higher salt content compared to the control group. Predictions of salinity at the initial sampling date were successful, with errors below 19 mg pot$^{-1}$ and $R^2$ values greater than 0.7 or 0.82, respectively. These predictions were mainly influenced by the variables GC and FC in the models (S4 Table). The results indicate that GC had an increasing effect on salt content in both experiments. However, this effect was not observed for FC, as increasing effects were only detected in Experiment 1 (S3 Fig).

**Inorganic N.**   The initial $N_{min}$-N contents in Exp. 1 ranged from 11 to 30 mg g$^{-1}$ N. Mixtures with a high FC share exhibited the highest $N_{min}$-N contents after four DaS. Similar results were observed in Exp. 2, where $N_{min}$-N contents ranged between 8.51 and 27.59 in the test mixtures.

By the end of the experiments, the $N_{min}$-N contents in some mixtures decreased to below 1 mg g$^{-1}$ N pot. The $N_{min}$-N contents were accurately predicted at all sampling dates (S2 Table) in both experiments. Initially, FC had the highest influence, while the GC and FC were identified as the most important factors influencing $N_{min}$-N immobilization at the end of the experiment, as indicated by the trace plot (S3 Fig).

Interestingly, the $NH_4$-N content in the control group was higher compared to the peat-reduced mixtures (S2 Table). Additionally, both RF and SF were found to influence $N_{min}$-N immobilization (S4 Table).

**Bulk density and plot volume.**   Pot density exhibited wide range among the tested mixtures in both experiments. The mixtures with a share of >34% v/v GC were estimated to have the highest pot density. Pot density could be accurately predicted with a high degree of accuracy, as indicated by an $R^2$ value greater than 0.78 and a RMSE of 0.03 g cm$^{-1}$, at all measurement points (S1 Table). The lowest densities were observed in the control group, as well as mixtures M4 and M5 in Exp. 1, and mixtures M4, M5, and M7 in Exp. 2.

According to the regression model (S4 Table) and the trace plot (S4 Fig), pot density was positively influenced by GC and negatively influenced by SF.

In general, it was observed that the test mixtures with high GC contents had slightly lower volumes in Exp. 1 However, this trend was not repeated in Exp. 2. The volume could not be accurately represented using a regression model (S3 Table).

**Stability.**   In terms of physical stability, the test mixtures in Exp. 1 exhibited values ranging from 0.39 to 0.77 kg cm$^{-2}$, while in Exp. 2, the values ranged between 0.41 and 0.54 kg cm$^{-2}$ (S1 Table). It appears that test mixtures with a high GC content tend to result in slightly higher stability values in both experiments (S1 Table).

The model quality for stability, as indicated by the RPD, was 1.53 in Exp. 1, allowing for a distinction between high and low stability values. The model parameters GC, FC, and SF had

the highest impact on the stability prediction (S3 Table). The trace plot revealed that stability is positively influenced by GC and negatively influenced by RF (S4 Fig).

**Water holding capacity.** In Exp. 1, the $WHC_{max}$ ranged from 42 to 52 g of water per pot initially, and decreased to a range of 36.09 g to 50.48 g of water at the end of the experiment. In Exp. 2, the highest $WHC_{max}$ was observed in the control group at the beginning of the experiment, while at the end of the experiment, the values ranged between 45.11 and 53.93 g of water in the test mixtures (S1 Table).

The $WHC_{max}$ was better predicted in Experiment 1 compared to Experiment 2 (S3 Table), indicating that the prediction quality for Experiment 2 was not sufficient for adequate prediction of WHC.

## Response surface method

Figs 2 and 3 depict the desirability of various decision combinations based on Table 2. These desirability values were calculated using parameters such as pH (at DaS 4), salt content (at DaS

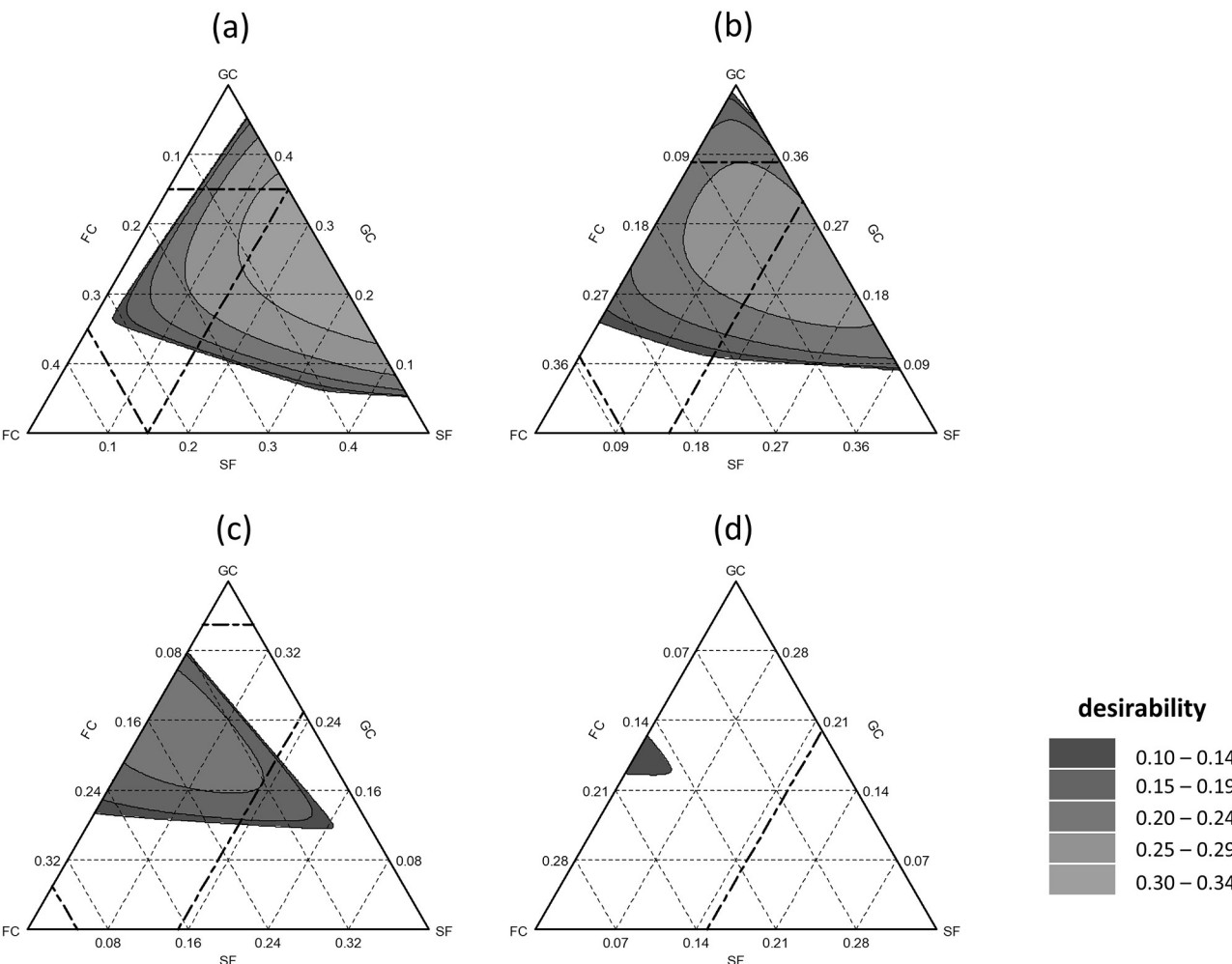

**Fig 2. Surface response of desirability for 50% v/v peat mixtures for green compost (GC), fermented compost fiber (FC), soft fiber (SF) and raw fiber (RF) in v/v.** Bold dashed lines are the mixture areas of peat substitutes tested. Were (a) = 0 v/v RF, (b) = 0.05 v/v RF, (c) = 0.1 v/v RF and (d) = 0.15 v/v RF.

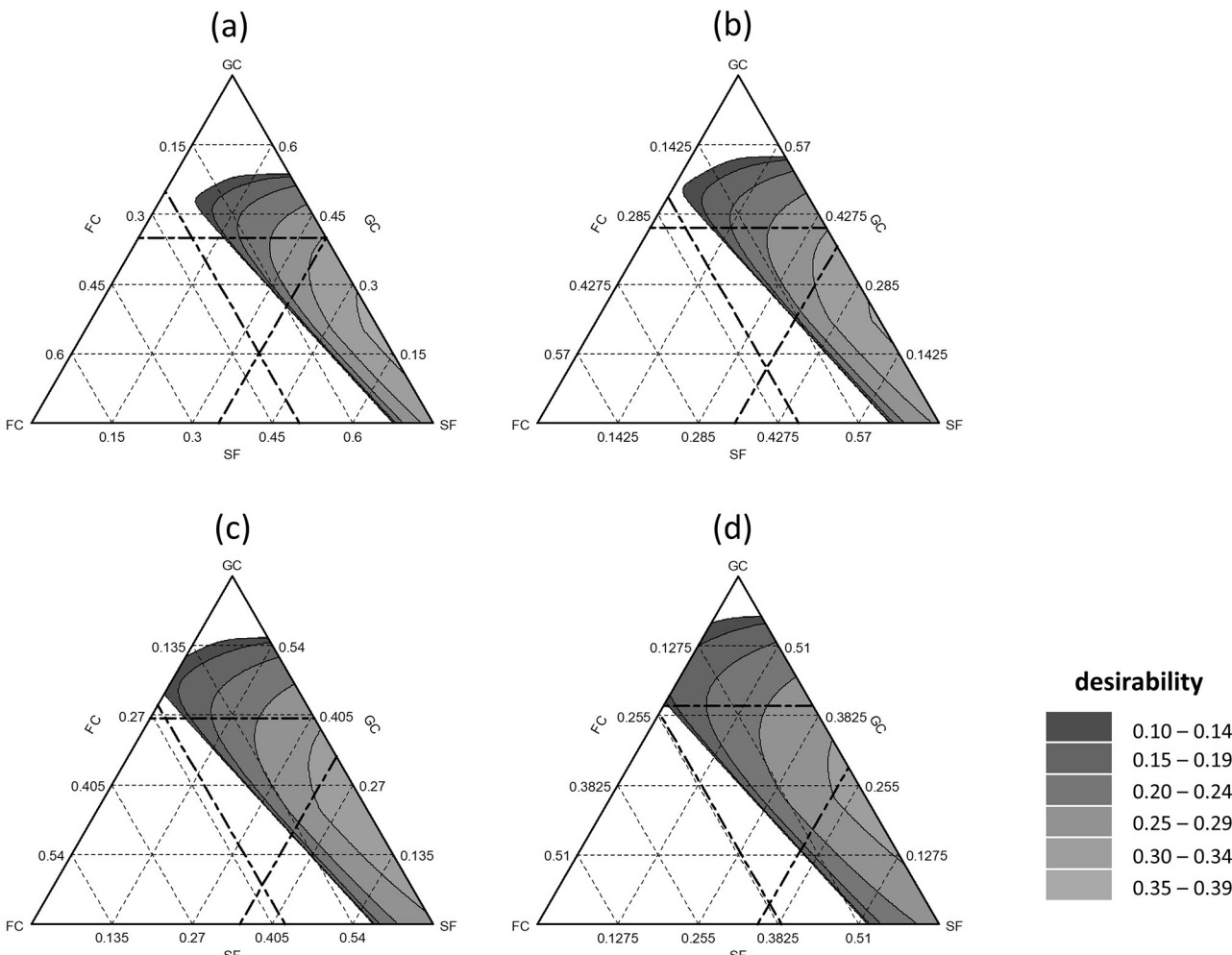

**Fig 3. Surface response of desirability for 25% v/v peat mixtures for green compost (GC), fermented compost fiber (FC), soft fiber (SF) and raw fiber (RF) in v/v.** Bold dashed lines are the mixture areas of peat substitutes tested. Were (a) = 0 v/v RF, (b) = 0.0375 v/v RF, (c) = 0.075 v/v RF and (d) = 0.1125 v/v RF.

4), $N_{min}$-N reduction (end of Experiment), $WHC_{max}$ (at DaS 4), physical stability (end of Experiment), and density (at DaS 4). However, it is important to note that these predictions mainly apply to the studied region and may not be reliably applicable to regions outside of the study area.

For Exp. 1, the mixtures with the highest desirability (GC: 22.5–35% v/v, FC: 0–12.5% v/v, SF: 10%-15%, and RF = 0) fell within the range of highest desirability ($\geq 0.3$) in the studied region (Fig 2a). When considering an RF of 5% v/v, the recommended range shifted to GC: 16–35% v/v, FC: 0–17.5% v/v, and SF: 2.5–15% v/v, with the optimal range being achieved, but with no desirability $>0.3$ (Fig 2b). Increasing the RF component to 10% v/v further reduced the area of desirability, with desirability values higher than 0.2 occurring within the range of GC: 15–27% v/v, FC: 5–29.5% v/v, and SF: 5–15% v/v (Fig 2c). Mixing shares of RF higher than 15% could not be recommended, as the recommended range covered only a small area with low desirability values (Fig 2d).

**Table 5. Examples of recommended mixtures for 50% v/v und 25% v/v peat mixtures.**

| Mixture constituents | GC [%] | FC [%] | SF [%] | RF [%] | Peat [%] | desirability |
|---|---|---|---|---|---|---|
| 3 | 35 | 0 | 15 | 0 | 50 | $\geq 0.3$ |
| 4 | 22.5–35 | 0–12.5 | 2–15 | 0 | 50 | $\geq 0.3$ |
| 5 | 16–35 | 0–17.5 | 2.5–15 | 5 | 50 | $\geq 0.25$ |
| 3 | 41 | 0 | 34 | 0 | 25 | $\geq 0.3$ |
| 4 | 32.25–39.75 | 0–7.5 | 33–34.5 | 0 | 25 | $\geq 0.3$ |
| 5 | 23.5–35.25 | 0–6 | 28.5–34.5 | 11.25 | 25 | $\geq 0.3$ |

In Exp. 2, the optimal range within the region with RF = 0 was GC: 32.5–39.75% v/v, FC: 0.75–7.5% v/v, and SF: 33–34.5% v/v (Fig 3a). At an RF of 7.5% v/v, a desirability of 0.3 or higher was achieved with GC: 24.75–39% v/v, FC: 0–8.5% v/v, and SF: 28.5–34.5% v/v (Fig 3). Table 5 presents exemplary recommended mixtures consisting of 3–5 components.

## Discussion

### Properties of growing media constituents

In this work, mixtures of five growing media constituents were used, not considering peat content as a fixed factor. GC and FC, as microbial converted organic matter, and wood fiber in different qualities (RF and SF) were used to assess different availabilities of one group component at a time. Although the recommended mixture uses all constituents or mixtures with 3 to 4 composts are possible. To avoid autocorrelations in the linear evaluation, relevant differences in the starting material should be considered.

The pH-value of the GC used in this experiment was lower than that of the compost used in other studies, which was above pH 8 in each case. This low pH appears to be beneficial for achieving the target values listed in Table 4, the pH value of the mixture can be raised due to the addition of compost [36]. Demonstrating the need to consider the quality of the initial substrates when substituting peat e.g. implementing a testing protocol to identify the best compost quality [37]. This is evidenced, for example, by the fact that admixtures of > 40% v/v GC did not cause a pH increase > 7, as reported by Herrera et al. [38]. Moreover, pH may decrease due to leaching of ions or uptake by the seedlings, which might be beneficial for plant nutrition in the last growing period. Fermented fiber materials typically have a low pH due to H+ ions accumulated while degrading cellulose [8].

The initial substrates FC and GC demonstrated high conductivity, however this was not a concern for the mixtures selected for this experiment [39]. It was recommended in practice to not exceed an admixture of 50% v/v compost with low salt content [12], which was verified in Exp. 2 for GC. During Exp. 1, the high salinity levels limited the addition of GC and FC to 35% each. To reduce the pH of the substrate mixture, the addition of wood fibers is suitable due to their low salt content.

The two wood fibers had high C/N ratios accordingly, which initially led to the decision of a general limitation of the fiber to < 15% v/v in the mixture with 50% v/v peat in Exp. 1 due to a high N immobilization potential. While this value is far from the maximum, a 30% v/v admixture was recommended by other scholars [13, 18]. However, different qualities of wood fiber in the mixture starting at 15% v/v can have a negative impact on plant nutrient supply [18]. Another possible reason could be a change in the binding capacity of the substrate, which can be decreased with increasing fiber fraction.

It is noteworthy that the substrate starting materials used in this work (acquired in different batches) did not exhibit important differences, indicating good management in compost production, particularly for GC and FC. The inclusion of the results from the experiment in the definition of limits for the subsequent experiment would be advantageous. This helps to extrapolate the results from Exp. 1 to Exp. 2. This could be attributed to the fact that the substrates are designed to meet the high requirements of industrial horticulture by the substrate industry.

## Seedling growth

The definition of maximum blend in the experiments were designed to prevent any detrimental effect on young plant growth and to focus on chemical and physical parameters to achieve the highest possible substrate quality. Both trials were successful, with no noteworthy differences to the control. However, the lower biomass in some variants compared to the control had no effect on the model in the regression analysis. The discrepancies in fresh mass between the trials can be attributed to the higher temperatures in Exp. 2, which were around 3.2˚C higher than in Exp. 1 after day seven (S1 and S2 Figs). This likely led to an increased growth rate. The biomasses had values similar to those reported in the literature [40, 41]. This indicates that, given the parameters chosen for the experiments, the risk of plant growth impairment was kept low. This provides greater flexibility in determining the optimal peat-reduced mixtures based on physical and chemical parameters and therefore, better fine-tuning. While under certain conditions, the fiber content may lead to N-immobilization in later growth stages, we have not investigated the effect on biomass after transplanting to the field.

## Physical and chemical properties of the test mixtures

As the pH value in all test mixtures of both experiments was below 7, the conditions for the seedlings in terms of pH value were in the optimal range for seedling cultivation for vegetables [7]. An increase in pH with increasing GC contents was expected and reflects previous experiments [22, 42]. The addition of FC lowered the pH, which in turn can be explained by the low initial pH caused by the organic acids produced during the fermentation process. Between DaS 4 and the end of the experiment, there was a small increase in pH in some mixtures, especially in Exp. 2 with 25% peat content. This could be related to the rapid reduction of $NH_4$-N in substrates with high compost contents (S2 Table). Due to a possibly higher proportion of microbial biomass with lower relative peat contents, N dynamics can be faster in those mixtures [43]. Next to that, the reduced buffering capacity of the wood fiber dominated mixtures also influence pH-value.

Similar to pH, conductivity showed the highest contents in the mixtures used with high proportions of GC and FC. However, irrigation during the cultivation period probably washed-out part of the salt, which was accompanied by a relevant reduction of the salinity. High salinity is one of the main problems in the use of GC [44]. Nonetheless, this is not a limitation factor in the present study, given that the conductivity in GC and FC are at a relatively low level and thus could be admixed to at least 30% [45, 46]. Admixture of GC and FC seems to be straightforward up to a level of about 40% if the conductivity in the starting material is > 3mS dm$^{-1}$. Although Raviv [11] showed that the proportion of compost should not exceed 50% v/v, effects of higher proportions could also be considered. Together with FC, which has a similar salinity as GC, a combined proportion of about 65% v/v in the mixture is also possible without negative effects on seedling growth. This is consistent with the results of Veeken et al. [47] and could be based on the content of wood fibers in the mixture, due to their low pH leading to dilution.

Generally, the results imply that a salinity of less than 205 mg pot$^{-1}$ is not detrimental to the cultivation of Chinese cabbage. Aligns with the research of who reported that Chinese cabbage seedlings can endure salinity levels of 2.9 g L$^{-1}$ (approximately 210 mg pot$^{-1}$) at pH 7. However, when both alkaline pH and high salinity are present, the plant growth of Chinese cabbage is adversely affected due to alkaline stress. Consequently, both pH and salinity should be taken into consideration.

The initial concentrations of N$_{min}$-N in the substrates used in the practical experiments were high, which meant that no additional fertilization was needed in Exp. 1. In some of the test mixtures, the content of N$_{min}$-N decreased to levels greater than 0.5 mg pot$^{-1}$ during the experiments, which could potentially lead to nutrient deficiency when used in the field. However, the mineralization of other nutrients from the organic components of the crop can also be expected. The lower reduction of N$_{min}$-N in mixtures with high GC content compared to those with high SF content may be related to the comparatively lower C/N ratios. GC showed a C/N ratio of less than 15, which likely did not lead to relevant N immobilization since the compost was already highly decomposed. It can be assumed that immobilization of nitrogen (N) occurs in the case of FC, given the C to N$_{org}$ (organic N) ratio of 35, due to the microbial availability of carbon compounds from the wood fibers, primarily from the long-chain compounds [48, 49]. Even with a fiber content of 40% v/v, an additional fertilization of 10 mg N did not lead to nutrient deficiencies in this experiment, meaning that N reduction is not a risk for seedling establishment. However, a further decrease in peat content would likely require additional fertilization. Nevertheless, the immobilized N could be re-mineralized later and thus should be available to the crop.

The relative importance of pot density for transport costs is clear, and substitution of peat with GC leads to increased pot weight. This is due to the higher bulk density of GC compared to peat, as discussed by Barrett et al. [12]. Although the bulk density of the mixture was higher than the 0.4 g cm$^{-3}$ value recommended by Abad et al. [50], this did not seem to affect plant growth suggest that a density above 0.4 g cm$^{-3}$ is acceptable, given the typically high mineral content of GC.

Generally, in practice pot volume is of lower importance in potted crop cultivation. However, the smaller the pot volume, the more important the relative volume differences becomes. In the small volume of press pots slight variations in pot height can result in a relative large additional material use. The pot volume differences between the two experiments in our research are mainly based on technical variations of the press pot machine. The variations between the mixtures of the individual tests can be attributed to the different physical properties, such as particle composition or press stability.

The WHC$_{max}$ of the mixtures varied, but the differences were minor, indicating that the suggested additional water requirement due to the use of press-stopped substrates by Gruda and Schnitzler [51] is not necessary. The higher water-holding capacities in the experiments were a result of the combination of different peat substitutes plus the addition of fibrous materials, which counteracted any negative effects of GC on water-holding capacity, according to Paradelo et al. [52]. Nevertheless, the results may not be universally applicable since the quality of GC can cause an increase or decrease in WHC$_{max}$ [53]. Furthermore, for press pots, physical stability is essential for successful machine planting; hence, high stability should be ensured.

The findings of this study indicate that the compost used here has a beneficial effect on physical stability. Conversely, a decrease in the amount of peat commonly leads to an augmentation in stability. Laun et al. [54] demonstrated that, up to a 50% v/v reduction of peat, stability was not affected. This may be attributed to the proportion of fine organic matter in the compost that contributes to the binding of the constituent parts.

## Decision-making and surface response methodology

The integration of the desirability calculation with the surface response method offers the advantage of being able to reduce the peat content in horticultural substrates in a practical manner without compromising the quality of the substrates. This is mainly due to the specified limits of the mixtures added. Therefore, Pascual et al. [7] requested transfer of laboratory analyses to horticultural practice can be achieved in a more precise way. The specifications for desirability (Table 4) are individual and can vary greatly depending on the intended use, making comparison with other sources difficult. This could be rectified by providing the model parameters (S3 and S4 Tables). The quality limits employed herein reflect the requirements described by Pascual et al. [7]. The inclusion of four mixtures has the benefit of allowing decisions on suitable mixtures to be made on the basis of two or three mixture components. This enables quick reactions to changes in the availability of peat substitutes, which can be highly variable in the market [1]. Since the components of mixtures are always interdependent [23], an overall view of all components that are mixed together should be sought. This is where a mixture design can be advantageous. In addition, the approach used here is particularly beneficial as it enables the reduction of peat content, thus making an important contribution to peat reduction in horticulture.

In many experiments, consideration of the entire mixture region is limited due to practical reasons [26, 55]. This is because it can lead to unusable results [55]. As the goal of mixing designs is to create a product that meets the requirements of the product users, the selected regions should be chosen in order to use the least amount of test mixtures to obtain meaningful results. Simplex centroid designs, such as those used by Ceglie el al [21] and Moldes et al. [56], have their limits in this regard. Therefore, computer generated designs are often employed to provide a more flexible approach [23, 24]. In contrast to the current study [28], employed an approach based on the simplex centroid to select the boundaries of the mixture components. Their results indicated that partial consideration was effective in identifying the optimal mixtures. The XVERT algorithm is advantageous due to its flexibility in boundary selection [24, 57]. When using a sequential approach, the experimental region is often no longer a simplex; this issue can be resolved through computer-generated mixture designs [55]. The findings from Exp. 1 were incorporated into Exp. 2, which additionally included the area outside of the examined region. As for black peat, which has the capacity to absorb and release nutrients equally [12], likely has little interaction with the other components in the mixture and thus the results of Exp. 1 can be applied to Exp. 2.

## Reduction of peat using mixture design and surface response

With an experimental approach using a mixture design, it becomes possible to gain valuable insights into the behavior of mixtures [23]. Such a design can also be used in combination with desirability to serve as a predictive tool. Notably, the components of a mixture are not independent of each other, making it hard to generalize results from experiments that lack a mixture design. Earlier studies have mostly employed methods that compared experimental mixtures employing ANOVA followed by a post hoc test [8, 17, 58]. This makes it challenging to make generalized statements about the potential of individual substitutes for peat substitution. A regression analysis approach appears to be more suitable for this purpose as shown in our study.

Our approach can clearly show the effects of peat substitute combination on substrate quality and it can forecast the optimal type and number of components in peat-reduced substrate mixes. This is in agreement with Pascual et al. [7], reporting a great potential transferring the

scientific approach of mixture design based on prediction model and surface response method to the horticultural industry.

The risk of unfavorable qualities for the peat industry could be reduced by gradually decreasing the proportion of peat in mixtures. A substantial reduction of peat in the young plant sector could be achieved at 25% v/v in the mixture. This would allow for fixed benchmarks to be set during the planning of trials. A further reduction of peat content to 0% may be possible by mainly utilizing GC and SF in the Exp. 2 mixtures; however, clay would need to be added to the mixtures to ensure the press ability of the substrates [59].

## Conclusion

The use of peat in horticulture can potentially be reduced by transferring scientific results to the peat industry. Our experimental work, conducted under conditions similar to those in horticultural practice, suggests that a targeted mixture design and the Respond Surface Method can be beneficial for the development of peat-reduced substrates. In particular, focusing on a defined mixing range could improve the efficiency of peat-reduced substrate development, as existing knowledge can be taken into account in the experimental design. Additionally, predictive models could be utilized to respond to changing seedling requirements. This approach could help the decision-making in the substrate industry, while also saving costs through a reduced experimental approach with few test mixtures. Further research is needed to assess whether the experimental approach employed in this study can be extended to other cultivars and branches of horticulture, particularly ornamental horticulture with its diverse and high demands.

## Supporting information

**S1 Fig. Air temperature (˚C) during Exp. 1 (˚C) in Exp. 1 (50% peat v/v).**
(PDF)

**S2 Fig. Air temperature (˚C) during Exp. 2 (25% peat v/v).**
(PDF)

**S3 Fig. Predicted response trace plot with piepels directions (based on peat free share) for pH-value, salt content and inorganic N immobilization for Exp. 1 and Exp. 2.**
(PDF)

**S4 Fig. Predicted response trace plot with piepels directions (based on peat free share) for WHC$_{max}$, physical stability and pot density for Exp.1 and Exp.2.**
(PDF)

**S1 Table. Mean, standard deviation (in parentheses) and critical difference (CD) for fresh plant biomass, stability, WHC$_{max}$, pot density, pH physical stability and pot volume of the substrate mixtures (M) and the peat control (C) after different time steps (DaS) of Exp. 1 (50% peat v/v) and Exp. 2 (25% peat v/v).**
(PDF)

**S2 Table. Mean, standard deviation (in parentheses) and critical difference (CD) for inorganic N (Nmin-N), ammonium (NH$_4$-N), stability, reduction of inorganic N and salt content of the substrate mixtures (M) and the peat control (C) after different time steps (DaS) of Exp. 1 (50% peat v/v) and Exp. 2 (25% peat v/v).**
(PDF)

**S3 Table. Estimates (based on peat free share) and prediction quality of linear predictions models and regression analysis for plant biomass, $WHC_{max}$, pot density, pH-value, stability and pot volume after different time steps (DaS) of Exp. 1 (50% peat v/v) and Exp. 2 (25% peat v/v).** Root mean squared error (RMSE), corrected multiple R-squared ($R^2$), ratio of performance to deviation (RPD). Significance level of estimate: ** = P < 0.001; * = P < 0.05. (PDF)

**S4 Table. Estimates (based on peat free share) and prediction quality of linear predictions models and regression analysis for inorganic N ($N_{min}$-N), ammonium N ($NH_4$-N), reduction of inorganic N, stability, and salt content after different time steps (DaS) of Exp. 1 (50% peat v/v) and Exp. 2 (25% peat v/v).** Root mean squared error (RMSE), corrected multiple R-squared ($R^2$), ratio of performance to deviation (RPD). Significance level of estimate: ** = P < 0.001; * = P < 0.05. (PDF)

## Acknowledgments

We are grateful to Klasmann-Deilmann GmbH and Kekkilä-Brill Substrates GmbH for supplying the substrate comportments. We thank Kai Katroschan from the "Centre of Horticultural crop Production, Mecklenburg-Vorpommern Research Centre for Agriculture and Fisheries, Gülzow-Prüzen, Germany by providing their infrastructure.

## Author Contributions

**Conceptualization:** André Sradnick.

**Data curation:** André Sradnick.

**Formal analysis:** André Sradnick.

**Funding acquisition:** André Sradnick.

**Investigation:** André Sradnick, Marie Werner.

**Methodology:** André Sradnick.

**Project administration:** Oliver Körner.

**Software:** André Sradnick.

**Supervision:** André Sradnick, Oliver Körner.

**Validation:** André Sradnick, Marie Werner.

**Visualization:** André Sradnick.

**Writing – original draft:** André Sradnick, Oliver Körner.

**Writing – review & editing:** André Sradnick, Marie Werner, Oliver Körner.

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
