## [Decision Letter · Decision Letter 0]

3 May 2023

PONE-D-23-09146Make a choice: A rapid strategy for minimizing peat in horticultural press pots substrates using an extreme vertices design and surface response approach

PLOS ONE

Dear Dr. Sradnick,

Thank you for submitting your manuscript to PLOS ONE. After careful consideration, we feel that it has merit but does not fully meet PLOS ONE’s publication criteria as it currently stands. Therefore, we invite you to submit a revised version of the manuscript that addresses the points raised during the review process.

We look forward to receiving your revised manuscript.

Kind regards,

Randall P. Niedz

Academic Editor

PLOS ONE

Journal Requirements:

Additional Editor Comments:

Greater clarity and additional information are required. Because mixture designs, and in particular constrained mixture designs, are not commonly used it would be very helpful to the reader to provide additional detail.  I suggest the following -

1) The design matrices used.  Include in the materials section. Be very explicit when describing the designs - e.g., a 5-component mixture that included .... Maybe include a small table that provides the proportional ranges for each component.

2) ANOVAs for all measured responses. I did not see any ANOVAs.  Most of these would be in the supplement but the salient effects would be included in the manuscript. This might take some thought as time was included.  Time could have been added as a process variable, but because it was not maybe select the end timepoint for the ANOVAs presented in the text.

3) Describe the data and model quality diagnostics used in the materials section. E.g., how was data analyzed to determine if a transform was required?  These analyses would be in the Results section.

4) Because mixture effects are very different conceptually from amount effects, include trace plots (the standard way to illustrate mixture component effects) of the salient effects.  Was there nonlinear blending?  This is precisely what mixture effects can detect that factorials cannot.  Nonlinear blending determines whether synergies or antagonistic effects were detected.

5) I would tend to not use the term "extreme vertices" but rather constrained as it is more intuitive to the reader not as familiar with mixture concepts.

6) Restructure the text to include the above.

Reviewers' comments:

Reviewer's Responses to Questions

**Comments to the Author**

1. Is the manuscript technically sound, and do the data support the conclusions?

Reviewer #1: Yes

Reviewer #2: Partly

2. Has the statistical analysis been performed appropriately and rigorously? 

Reviewer #1: No

Reviewer #2: Yes

3. Have the authors made all data underlying the findings in their manuscript fully available?

Reviewer #1: Yes

Reviewer #2: Yes

4. Is the manuscript presented in an intelligible fashion and written in standard English?

Reviewer #1: Yes

Reviewer #2: No

5. Review Comments to the Author

Reviewer #1: I have completed my evaluation of the manuscript intituled: Make a choice: A rapid strategy for minimizing peat in horticultural press pots substrates using an extreme vertices design and surface response approach.

The work presented by the authors is very interesting.

I agree to the publication, and the justifications are:

The abstract is clearly presented and are related to the data presented in the results section. The introduction provides sufficient information about the subject addressed. Materials and methods are adequately described;

The overall value of the manuscript is excellent, but the presentation needs a detailed correction: In figures 2 and 3, the numbers inside the figure's white triangle are unclear. Please, edit the figure to show the numbers clearly.

Reviewer #2: I have reviewed the manuscript titled “Make a choice: A rapid strategy for minimizing peat in horticultural press pots substrates using an extreme vertices design and surface response approach”, and overall, I find the work to be promising. However, I have some concerns regarding the presentation of the results.

In my opinion, the results section is poorly structured, and it is challenging for the reader to follow the logical flow of the study. This is the biggest problem. The authors should carefully consider improving this section.

The captions for the tables and figures are not informative enough, and it is difficult to understand the data without having to go back to the main text.

Additionally, the manuscript uses several abbreviations and short forms for terminologies, which are not adequately explained or defined in the text (Section Experimental Design starting with line 139).

Furthermore, the naming of mixtures using the letter M is of course reasonable and should be implemented throughout the tables as well.

Overall, I believe that the manuscript has the potential to make a significant contribution to the field, and I recommend it for publication after the necessary revisions have been made.

6. PLOS authors have the option to publish the peer review history of their article (what does this mean?). If published, this will include your full peer review and any attached files.

Reviewer #1: No

Reviewer #2: No

---

## [Author Response · Author response to Decision Letter 0]

17 Jun 2023

PONE-D-23-09146

Make a choice: A rapid strategy for minimizing peat in horticultural press pots substrates using an extreme vertices design and surface response approach

Additional Editor Comments:

Greater clarity and additional information are required. Because mixture designs, and in particular constrained mixture designs, are not commonly used it would be very helpful to the reader to provide additional detail. I suggest the following -

1) The design matrices used. Include in the materials section. Be very explicit when describing the designs - e.g., a 5-component mixture that included .... Maybe include a small table that provides the proportional ranges for each component.

Response: We have revised the materials and methods part intensively and included two tables (Table 2 and Table 3) for better presentation of the methods (line 102-187). 

2) ANOVAs for all measured responses. I did not see any ANOVAs. Most of these would be in the supplement but the salient effects would be included in the manuscript.

Response: Thank you for the advice. We have been added the CD (critical distance: confidence interval from TukeyHSD / 2) after ANOVA in the supplement to prevent clarity of the tables (please look at Tables S1 table 1 and S2 table 2) . Several adaptions are made in results part too (line 192-283).

 This might take some thought as time was included. Time could have been added as a process variable, but because it was not maybe select the end timepoint for the ANOVAs presented in the text.

Response: Apologies for the discrepancy in the results section where it was mistakenly stated that the time effect was determined through a test. Due to the nature of the experimental design, the time steps were not independent. To ensure clarity, we have made intensive revisions to all parts of the results section (line 197-294).

3) Describe the data and model quality diagnostics used in the materials section. E.g., how was data analyzed to determine if a transform was required? 

Response: We have been testing the all-data for homogeneity of variances by three means: Levene’s test, Shapiro-Wilk test and Durbin-Watson test. To our satisfaction (and indeed somewhat surprisingly) all datasets have passed all three tests. We have thoroughly double-checked the data and R-coding, and it is all correct. Due that and according to the statistical rules no adaption was required. Additional information was added in line: 164-167.

These analyses would be in the Results section.

Response: The results section was extensively revised (line 197-294)

4) Because mixture effects are very different conceptually from amount effects, include trace plots (the standard way to illustrate mixture component effects) of the salient effects.

Response: Thanks for this helpful hint, we have determined for the main datasets "Piepel direktions" and added them to the supplementary (S7-S8).

 Was there nonlinear blending? This is precisely what mixture effects can detect that factorials cannot. Nonlinear blending determines whether synergies or antagonistic effects were detected.

Response: Thank you for this important note. In addition to the linear model, the data sets were also tested for quadratic relationships. Here, interaction effects were mostly very weak. Likewise, the variance inflation factor (VIF) increased to such an extent that the uncertainty for the parameters shown is too pronounced.

5) I would tend to not use the term "extreme vertices" but rather constrained as it is more intuitive to the reader not as familiar with mixture concepts.

Response: Thank you for the hint. “Extreme vertexes design” is converted to:“constrained mixture design”

6) Restructure the text to include the above.

Reviewer #1: I have completed my evaluation of the manuscript intituled: Make a choice: A rapid strategy for minimizing peat in horticultural press pots substrates using an extreme vertices design and surface response approach.

The work presented by the authors is very interesting.

I agree to the publication, and the justifications are:

The abstract is clearly presented and are related to the data presented in the results section. The introduction provides sufficient information about the subject addressed. Materials and methods are adequately described;

The overall value of the manuscript is excellent, but the presentation needs a detailed correction: In figures 2 and 3, the numbers inside the figure's white triangle are unclear. Please, edit the figure to show the numbers clearly.

Response: We have modified the figures in order to make the displayed numbers in the illustrations more visible (see Fig2 and Fig3).

Reviewer #2: I have reviewed the manuscript titled “Make a choice: A rapid strategy for minimizing peat in horticultural press pots substrates using an extreme vertices design and surface response approach”, and overall, I find the work to be promising. However, I have some concerns regarding the presentation of the results.

In my opinion, the results section is poorly structured, and it is challenging for the reader to follow the logical flow of the study. This is the biggest problem. The authors should carefully consider improving this section.

Response: Thank you very much for the comment. The results section has been intensively edited and strongly rewritten to make the measurement data clearer. Line: 198-298.

The captions for the tables and figures are not informative enough, and it is difficult to understand the data without having to go back to the main text.

All table and figure captions have been revised to facilitate interpretation of the results (see, line 633-656, Table 1-5 and line 291-298).

Additionally, the manuscript uses several abbreviations and short forms for terminologies, which are not adequately explained or defined in the text (Section Experimental Design starting with line 139).

Response: Thank you very much for the hint. The text has been revised in line with your comment (line 106-109).

Furthermore, the naming of mixtures using the letter M is of course reasonable and should be implemented throughout the tables as well.

Response: The tables in the supplementary now have the designation M1-MX. Table2 and table S1-S2

Overall, I believe that the manuscript has the potential to make a significant contribution to the field, and I recommend it for publication after the necessary revisions have been made.________________________________________

---

## [Editor Report · Decision Letter 1]

17 Jul 2023

Make a choice: A rapid strategy for minimizing peat in horticultural press pots substrates using a constrained mixture design and surface response approach

PONE-D-23-09146R1

Dear Dr. Sradnick,

We’re pleased to inform you that your manuscript has been judged scientifically suitable for publication and will be formally accepted for publication once it meets all outstanding technical requirements.

Kind regards,

Randall P. Niedz

Academic Editor

PLOS ONE
---

## [Editor Report · Acceptance letter]

21 Jul 2023

PONE-D-23-09146R1 

Make a choice: A rapid strategy for minimizing peat in horticultural press pots substrates using a constrained mixture design and surface response approach 

Dear Dr. Sradnick:

I'm pleased to inform you that your manuscript has been deemed suitable for publication in PLOS ONE. Congratulations! Your manuscript is now with our production department. 

Kind regards, 

on behalf of

Dr. Randall P. Niedz 

Academic Editor

PLOS ONE